# COVID-19 Signs and Symptom Clusters in Long-Term Care Facility Residents: Data from the GeroCovid Observational Study

Alba Malara [1,*], Marianna Noale [2], Angela Marie Abbatecola [3], Gilda Borselli [4], Carmine Cafariello [5], Stefano Fumagalli [6], Pietro Gareri [7], Enrico Mossello [6], Caterina Trevisan [8,9], Stefano Volpato [9], Fabio Monzani [10], Alessandra Coin [8], Giuseppe Bellelli [11], Chukwuma Okoye [10], Stefania Del Signore [12], Gianluca Zia [12], Raffaele Antonelli Incalzi [13] and on behalf of the GeroCovid LTCFs Working Group [†]

1   ANASTE-Humanitas Foundation, Via dei Gracchi 137, 00192 Rome, Italy
2   Aging Branch, Neuroscience Institute, National Research Council, Via Giustiniani 2, 35128 Padua, Italy; marianna.noale@in.cnr.it
3   Alzheimer's Disease Clinic Department, Azienda Sanitaria Locale (ASL), Via Colle Melfa 75, 03042 Atina, Italy; angela_abbatecola@yahoo.com
4   Italian Society of Gerontology and Geriatrics (SIGG), Via G.C. Vanini 5, 50129 Florence, Italy; gilda.borselli@sigg.it
5   Geriatrics Outpatient Clinic and Territorial Residences, Italian Hospital Group, Via Tiburtina 188, 00012 Guidonia, Italy; ccafariello@italianhospitalgroup.com
6   Department of Experimental, Clinical Medicine, Division of Geriatric and Intensive Care Medicine, University of Florence and AOU, Largo Brambilla 3, 50134 Florence, Italy; stefano.fumagalli@unifi.it (S.F.); enrico.mossello@unifi.it (E.M.)
7   Center for Cognitive Disorders and Dementia (CDCD) Catanzaro Lido–ASP Catanzaro 214, 88100 Catanzaro Lido, Italy; pietro.gareri@me.com
8   Geriatric Division, Department of Medicine (DIMED), University of Padua, Via Giustiniani 2, 35128 Padua, Italy; caterina.trevisan.5@studenti.unipd.it (C.T.); alessandra.coin@unipd.it (A.C.)
9   Geriatric and Orthogeriatric Division, Department of Medical Science, University of Ferrara, Via Aldo Moro 2, 44124 Cona, Italy; vlt@unife.it
10  Geriatrics Unit, Department of Clinical and Experimental Medicine, University of Pisa, Via Paradisa 2, 56124 Pisa, Italy; fabio.monzani@unipi.it (F.M.); chuma@hotmail.it (C.O.)
11  Acute Geriatric Unit, School of Medicine and Surgery, University of Milano-Bicocca, San Gerardo Hospital, Via G. B. Pergolesi 33, 20900 Monza, Italy; giuseppe.bellelli@unimib.it
12  Bluecompanion Ltd., 237 Vauxhall Bridge Rd, Pimlico, London SW1V 1EJ, UK; susanna.ds@bluecompanion.eu (S.D.S.); gianluca.zia@bluecompanion.eu (G.Z.)
13  Department of Internal Medicine and Geriatrics, Campus Bio-Medico University, Via Alvaro del Portillo 200, 00128 Rome, Italy; r.antonelli@unicampus.it
*   Correspondence: albamalara@gmail.com
†   The GeroCovid LTCFs Working Group members list is shown in the Acknowledgements.

**Abstract:** Background: Long-term care facility (LTCF) residents often present asymptomatic or paucisymptomatic features of SARS-CoV-2 infection. We aimed at investigating signs/symptoms, including their clustering on SARS-CoV-2 infection and mortality rates associated with SARS-CoV-2 infection in LTCF residents. Methods: This is a cohort study of 586 aged ≥ 60 year-old residents at risk of or affected with COVID-19 enrolled in the GeroCovid LTCF network. COVID-19 signs/symptom clusters were identified using cluster analysis. Cluster analyses associated with SARS-CoV-2 infection and mortality were evaluated using logistic regression and Cox proportional hazard models. Results: Cluster 1 symptoms (delirium, fever, low-grade fever, diarrhea, anorexia, cough, increased respiratory rate, sudden deterioration in health conditions, dyspnea, oxygen saturation, and weakness) affected 39.6% of residents and were associated with PCR swab positivity (OR = 7.21, 95%CI 4.78–10.80; $p < 0.001$). Cluster 1 symptoms were present in deceased COVID-19 residents. Cluster 2 (increased blood pressure, sphincter incontinence) and cluster 3 (new-onset cognitive impairment) affected 20% and 19.8% of residents, respectively. Cluster 3 symptoms were associated with increased mortality (HR = 5.41, 95%CI 1.56–18.8; $p = 0.008$), while those of Cluster 2 were not associated with mortality (HR = 0.82, 95%CI 0.26–2.56; $p = 730$). Conclusions: Our study highlights that delirium, fever, and

low-grade fever, alone or in clusters should be considered in identifying and predicting the prognosis of SARS-CoV-2 infection in older LTCF patients.

**Keywords:** COVID-19; long term care facilities; gerocovid observational study; symptoms cluster

## 1. Introduction

Clinical presentations of coronavirus disease-19 (COVID-19) can significantly vary from asymptomatic infection to severe respiratory failure [1]. In adults, common clinical manifestations of COVID-19 include nasal secretions, cough, dyspnea, fever, myalgia, and occasionally diarrhea. Approximately 15% have developed acute respiratory distress syndrome that may last from 5 to 14 days [2]. Reflecting that COVID-19 infection symptoms particularly vary in older LTCF adults, numerous atypical manifestations including delirium, falls, muscle wasting, anorexia, and cachexia have been shown to be associated with COVID-19 infection [3]. Therefore, the need to quickly recognize COVID-19 infection in these residents in order to protect against negative prognostic outcomes as well as rapidly reduce the spread of the infection in this setting remains crucial. Indeed, older residents commonly suffer from multiple comorbidities that may mimic SARS-CoV-2 infection, thus underlining clinical difficulties related to identifying COVID-19 in this setting. Recent literature has underlined that older residents with three or more chronic diseases, such as dementia or cognitive impairment, malnutrition, or central and peripheral arterial disease have a higher risk of an infection from SARS-CoV-2 [4]. Interestingly, these authors also found that mortality was significantly higher in SARS-CoV-2-positive residents than in SARS-CoV-2-negative residents with suspicious symptoms (21.6% vs. 10.8%, respectively) [4]. At the moment, implications and clinical relevance of asymptomatic and paucisymptomatic COVID-19 residents remain unclear as well as specific treatment options and type of clinical monitoring in LTCF residents [5]. The prevalence of asymptomatic cases greatly varies, from 16% to 69.7% in populations worldwide [6].

Even though anti-SARS-CoV-2 vaccines have shown to significantly lower mortality rates, infection rates in LTCFs remain high. It also is still unclear why clinical presentations of COVID-19 infection in older patients largely vary (asymptomatic, typical clinical symptoms or atypical symptoms), thus underlining an urgent need to identify which presentations may be significantly related to negative clinical outcomes. In this study, we aimed at identifying signs or symptoms, as well as the clustering of signs/symptoms, associated with a SARS-CoV-2 infection and evaluate the related risk on negative outcomes including mortality.

## 2. Materials and Methods

GeroCovid LTCFs is a part of the GeroCovid Observational Study, a multi-center and multi-setting study evaluating the impact of the COVID-19 pandemic on the health outcomes of older patients in numerous clinical settings of acute and long-term care [7,8].

### 2.1. Participants

The GeroCovid LTCF cohort included 39 sites from 6 Italian regions (n = 2380). For this study analysis, we included nine study sites that reported positive COVID-19 cases from 1 March 2020 to 31 December 2020. For study purposes, we included 586 residents aged ≥ 60 years with suspicious signs or symptoms or who were considered at a high risk of a COVID-19 infection.

Onset symptoms included: (i) "typical" (cough, nasal congestion, hoarseness, sore throat, wheezing, sneezing, loss of sense of smell or taste, or high temperature); (ii) "atypical" (diarrhea, vomiting, anorexia, delirium, weakness). High-risk contacts were defined as residents who had direct physical contact with a COVID-19 confirmed case or were in a closed environment with a COVID-19 confirmed case in the absence of suitable personal

protective equipment. Furthermore, all new residents admitted to the LTCF or readmitted after a period of hospitalization were considered at risk of infection. Residents with suspicious signs and/or symptoms or at risk of COVID-19 infection were isolated and tested for SARS-CoV-2 positivity using PCR–RNA testing. According to swab results, residents were then categorized as positive or negative for SARS-CoV-2 infection [4].

Mobility assessments over the last month were determined using data from the Frailty Anamnestic Criteria [9]. Low-grade fever was defined as a body temperature ranging from 37 °C to 37.5 °C, while high-grade fever was defined as a body temperature higher than 37.5 °C.

### 2.2. Measures and Data Collection

Data on sociodemographic variables, comorbidities, polypharmacy, mobility, symptoms, SARS-CoV-2 swab testing, and outcome (clinical course, transfer to a different setting, death) were collected.

The Campus Bio-Medico University Ethical Committee approved the overarching protocol of the GeroCovid Observational study on 3 April 2020 (Trial Registration: NCT04379440). All participating investigational sites gained approval from their local Ethical Committee review board. Informed consent was aquired and the data were collected using a national de-identified electronic registry provided by BlueCompanion.

### 2.3. Statistical Analysis

The clinical characteristics of the study participants are reported as means $\pm$ standard deviation (SD) or median [25–75th percentile] for quantitative measures and as counts or percentages for categorical variables. The normality of the distributions was evaluated using the Shapiro–Wilk test. Clinical characteristics were summarized and compared among groups (positive vs negative SARS-CoV-2 swab) using the chi-squared or Fisher's exact tests for the categorical variables and the generalized linear model or the Wilcoxon rank-sum test for the quantitative ones. A multivariable logistic regression model adjusted for age, sex, and comorbidity (defined as having three or more comorbidities, according to the median of the sample distribution) evaluated the correlation of having a positive swab with reported signs and symptoms. A stepwise analysis on symptoms was performed with a $p$-value of 0.15 to entry and a $p$-value 0.20 to be retained in the model. Tjur $R^2$ was calculated to evaluate the predictive power of the model [10]. The results are presented as adjusted odds ratios (OR) and 95% confidence intervals (95%CI).

The presence of clusters among COVID-19 symptoms was evaluated using a hierarchical cluster analysis (McQuitty method) as a similarity measure of the proportion of observations when two symptoms were simultaneously present. A dendrogram (a tree-like diagram that illustrates the relationships between symptoms according to the measure of similarity chosen) obtained from cluster analysis was evaluated. The horizontal axis represents the similarity between clusters, while the vertical axis represents the considered symptoms, and each joining of two clusters is represented by the splitting of a horizontal line into two horizontal lines. This analysis started with individual symptoms, and clusters of the most similar symptoms were progressively formed, joining symptoms and clusters until all symptoms were joined into a single large cluster. The association between each cluster and a positive SARS-CoV-2 swab was evaluated using logistic regression models adjusted for age and sex.

Cox proportional hazard models were performed to determine probability risk of short-term mortality, and independent covariates regarding age, sex, number of comorbidities, and COVID-19 positivity were included in models. Additional Cox models with symptom clusters as independent variables adjusted for age, sex, and number of comorbidities were also performed. The results are presented as adjusted hazard ratios (HR) and 95% confidence intervals (95%CI).

All statistical tests were two-tailed, and statistical significance was assumed for $p$-value < 0.05. The analyses were performed using SAS, V.9.4 (SAS Institute, Cary, NC, USA).

### 3. Results

The study included 586 residents (mean age 84.8 ± 8.5 years, range 60–100 years, 72.9% women) based on the presence of suspicious signs and/or symptoms of a SARS-CoV-2 infection and those at a high risk of a SARS-CoV-2 infection. SARS-CoV-2 RNA testing using RT-PCR was performed in 583 residents and identified 209 positive SARS-CoV-2 residents. As reported in Table 1, the use of polypharmacy (median number of seven drugs) and having three or more comorbidities were significantly higher in those with a SARS-CoV-2 infection compared with those without infection. Furthermore, mobility worsening in the last month was significantly higher in those with SARS-CoV-2 infection compared with those without (72% vs. 28%) ($p < 0.001$).

**Table 1.** Characteristics of older adults from the GeroCovid LTCFs study: overall population and by SARS-CoV-2 positive or negative swab results.

| | All (n = 586) | SARS-CoV-2 + (n = 209) | SARS-CoV-2 − (n = 374) | *p*-Value |
|---|---|---|---|---|
| Age, year, mean ± SD | 84.8 ± 8.5 | 85.5 ± 8.1 | 84.4 ± 8.6 | 0.19 |
| Sex, female, n (%) | 427 (72.9) | 152 (72.7) | 273 (73.0) | 0.94 |
| Smoking status, n (%) | | | | |
| Current smoker | 11 (3.8) | 1 (1.1) | 9 (4.6) | 0.08 |
| Ex-smoker | 46 (15.8) | 20 (21.3) | 26 (13.3) | |
| Non smoker | 234 (80.4) | 73 (77.7) | 160 (82.0) | |
| Number of drugs, median (Q1, Q3) | 5 (4, 7) | 7 (5, 10) | 5 (3, 7) | <0.001 |
| Total number of chronic diseases, median (Q1, Q3) (available for n = 594) | 3 (2, 5) | 3 (2, 4) | 2 (1, 4) | 0.002 |
| Chronic diseases, n (%) | | | | |
| 0, 1, 2 | 224 (38.2) | 57 (27.3) | 167 (44.7) | <0.001 |
| 3+ | 365 (61.8) | 152 (72.7) | 207 (55.3) | |
| Worsening of mobility in the last month, n (%) (available for n = 271 residents) | 116 (42.8) | 67 (72.0) | 49 (27.5) | <0.001 |

Abbreviations: SD, Standard Deviation; Q1, Quartile 1; Q3, Quartile 3.

An amount of 503 residents had full data regarding any signs and/or symptoms of infection. Of these, approximately 30% of SARS-CoV-2-positive residents did not report any symptoms, while over 70% reported at least one symptom (Table 2). The most common symptom in older residents with a SARS-CoV-2 infection was delirium (41.2%), followed by high-grade fever (39.1%), low-grade fever (36.2%), sudden worsening of health status (35%), weakness (32.1%), low oxygen saturation at rest (SpO2 < 90%) (29.6%), anorexia (27.0%), dyspnea (26.1%), diarrhea (21.6%), and diuresis contraction (14.2%) (Table 2). According to logistic regression analyses, clinical features associated with RT-PCR positivity were delirium (OR = 9.9; 95%CI: 3.5–27.5; $p < 0.001$), high-grade fever (OR = 7.0; 95%CI: 3.1–16.1; $p < 0.001$), low-grade fever (OR = 4.3; 95%CI: 1.5–12.2; $p = 0.006$), and having three or more comorbidities (OR = 2.0; 95%CI: 1.0–3.7; $p = 0.038$) (Figure 1). The prevalence of delirium as the onset symptom of a SARS-CoV-2 infection was significantly higher in residents with dementia compared with those without dementia (27.1% and 10.5%, respectively; $p = 0.001$).

**Table 2.** Symptoms of older adults from the GeroCovid LTCFs study, according to SARS-CoV-2 infection status.

| | SARS-CoV-2 + (n = 179) | SARS-CoV-2 − (n = 324) | *p*-Value |
|---|---|---|---|
| No symptoms, n (%) | 53 (29.6) | 203 (62.7) | <0.001 |
| At least one symptom, n (%) | 126 (70.4) | 121 (37.3) | <0.001 |
| Fever, n (%) | 70 (39.1) | 21 (6.5) | <0.001 |
| Low-grade fever, n (%) | 64 (36.2) | 8 (2.5) | <0.001 |
| Pharyngodynia, n (%) | 1 (1.3) | 3 (1.0) | 1.000 |
| Cough, n (%) | 21 (12.4) | 16 (5.0) | 0.003 |
| Sneezing, n (%) | 4 (2.3) | 6 (1.9) | 0.72 |
| Dyspnoea, n (%) | 46 (26.1) | 18 (5.7) | <0.001 |
| Low oxygen saturation after walking, n (%) | 2 (2.0) | 6 (2.1) | 1.000 |
| Low oxygen saturation at rest (<90%), n (%) | 37 (29.6) | 15 (4.9) | <0.001 |
| S 02 %, mean±SD | 95 (93, 96) | 97 (96, 98) | <0.001 |
| Weakness/Prostration, n (%) | 52 (32.1) | 41 (12.7) | <0.001 |
| Fall or fainted, n (%) | 1 (0.9) | 7 (2.3) | 0.69 |
| Muscles aching, n (%) | 10 (6.6) | 13 (4.1) | 0.24 |
| Delirium, n (%) | 49 (41.2) | 7 (2.3) | <0.001 |
| Conjunctivitis, n (%) | 3 (1.8) | 5 (1.6) | 1.000 |
| Loss of smell (if new), n (%) | 0 (0.0) | 0 (0.0) | − |
| Loss of taste, n (%) | 3 (2.2) | 2 (0.6) | 0.17 |
| Anorexia, n (%) | 30 (27.0) | 20 (6.6) | <0.001 |
| Nausea/vomiting, n (%) | 12 (7.2) | 4 (1.3) | 0.004 |
| Diarrhea, n (%) | 36 (21.6) | 12 (3.8) | <0.001 |
| Raynaud syndrome, n (%) | 4 (3.5) | 0 (0.0) | 0.005 |
| Cutaneous symptoms, n (%) | 6 (5.1) | 2 (0.7) | 0.007 |
| Sudden worsening of health status, n (%) | 43 (35.0) | 5 (1.6) | <0.001 |
| Aphasia/dysnomia, n (%) | 1 (1.0) | 6 (2.0) | 0.68 |
| Cognitive Impairment, n (%) | 27 (30.0) | 49 (16.3) | 0.004 |
| Diuresis contraction, n (%) | 17 (14.2) | 5 (1.6) | <0.001 |
| Urines of faeces incontinence, n (%) | 5 (4.5) | 47 (15.9) | 0.002 |
| Unable to ask questions, n (%) | 3 (4.4) | 24 (9.2) | 0.21 |
| Unable to fill a self-evaluation questionnaire, n (%) | 7 (10.6) | 35 (13.6) | 0.52 |
| Number of symptoms, median (Q1, Q3) | 2 (0, 6) | 0 (0, 2) | <0.001 |
| Number of symptoms, n (%) | | | |
| 0 | 53 (29.6) | 203 (62.7) | |
| 1 | 25 (14.0) | 36 (11.1) | <0.001 |
| 2+ | 101 (56.4) | 85 (26.2) | |

Abbreviations: Q1, Quartile 1; Q3, Quartile 3.

Cluster analysis identified three symptom clusters (Figure 2): Cluster 1, which included delirium, fever, low-grade fever, diarrhea, anorexia, cough, increased respiratory frequency, dyspnea, low oxygen saturation at rest, and weakness/prostration, was present in 39.6% of the residents; Cluster 2 included recent-onset incontinence, increased blood pressure, and the inability to fill a self-evaluation questionnaire and was present in 20% of the residents; Cluster 3 was defined by new-onset cognitive impairment and included 19.8% of residents. The percentage of residents for each cluster and the association with a positive or negative SARS-CoV-2 swab test are reported in Figure 3. Only Cluster 1 and Cluster 3 symptoms were significantly associated with an increased probability of having a positive PCR swab test (OR = 7.21, 95%CI: 4.8–10.8, *p* < 0.001; OR = 2.05, 95%CI: 1.18–3.56, *p* = 0.01, respectively), while symptoms in Cluster 2 did not correlate with a positive PCR test (OR = 0.75, 95%CI: 0.44–1.29, *p* = 0.295).

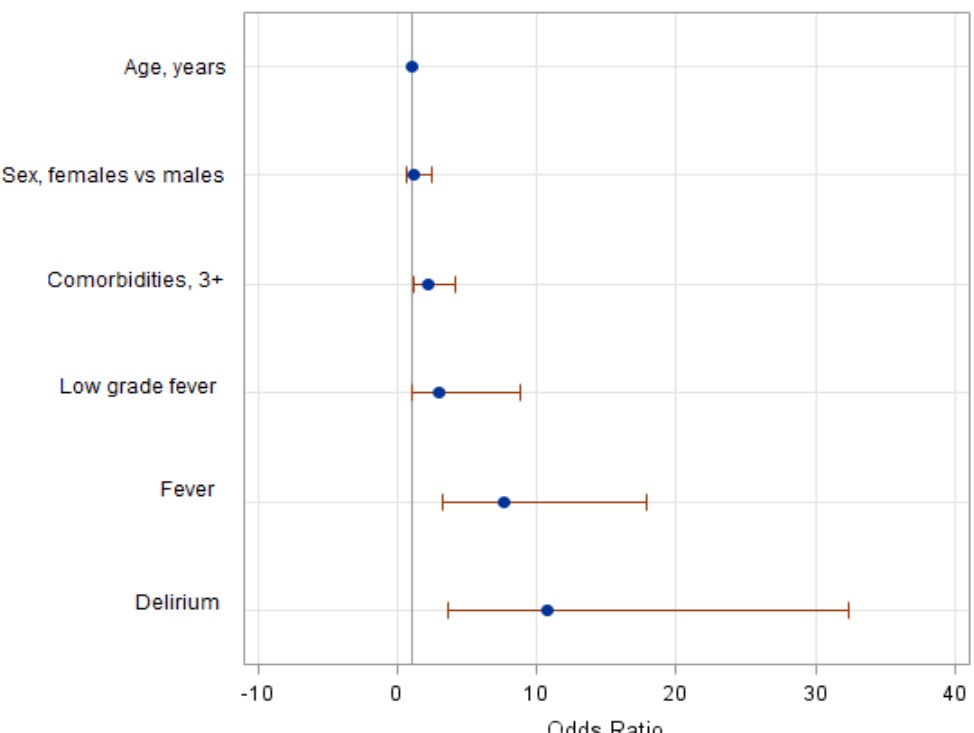

**Figure 1.** The associations between clinical features and SARS-CoV-2 infection (according to PCR swab testing). Logistic regression model using stepwise analysis (sle = 0.15; sls = 0.20), adjusted for age, sex, and comorbidity (defined as having 3+ chronic diseases). Symptoms reported by at least 5% of study participants (including fever, low-grade fever, cough, dyspnea, low oxygen saturation at rest, weakness/prostration, delirium, anorexia, diarrhea, sudden worsening of health status, diuresis contraction, urine or feces incontinence, inability to ask questions, inability to fill a self-evaluation questionnaire) were considered possible independent variables.

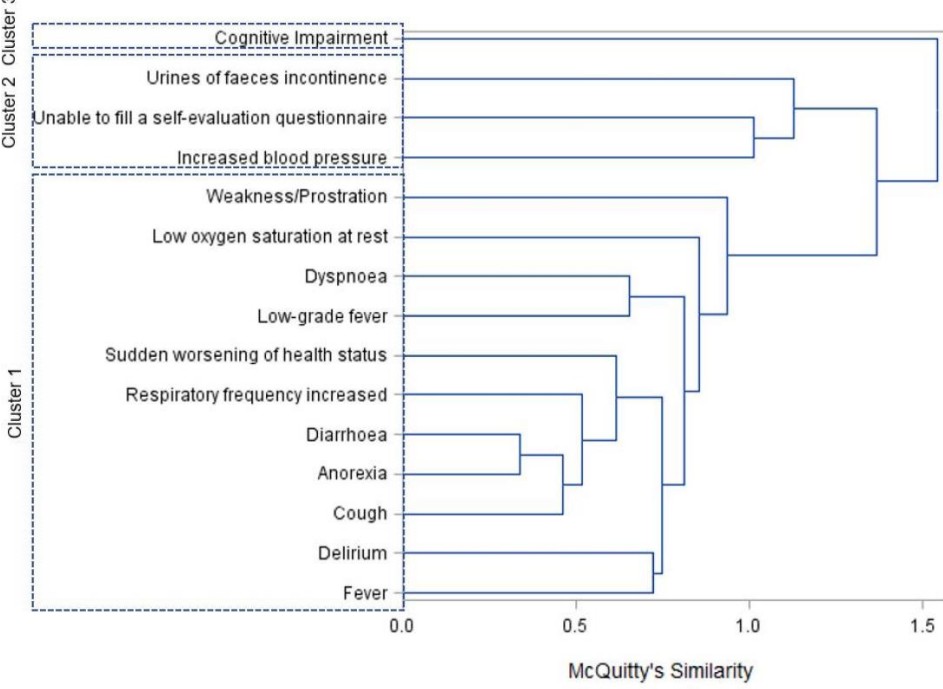

**Figure 2.** The dendrogram of symptom clusters (n = 503 participants independent of swab results; McQuitty method; only symptoms reported for 5% or more of the study participants were included).

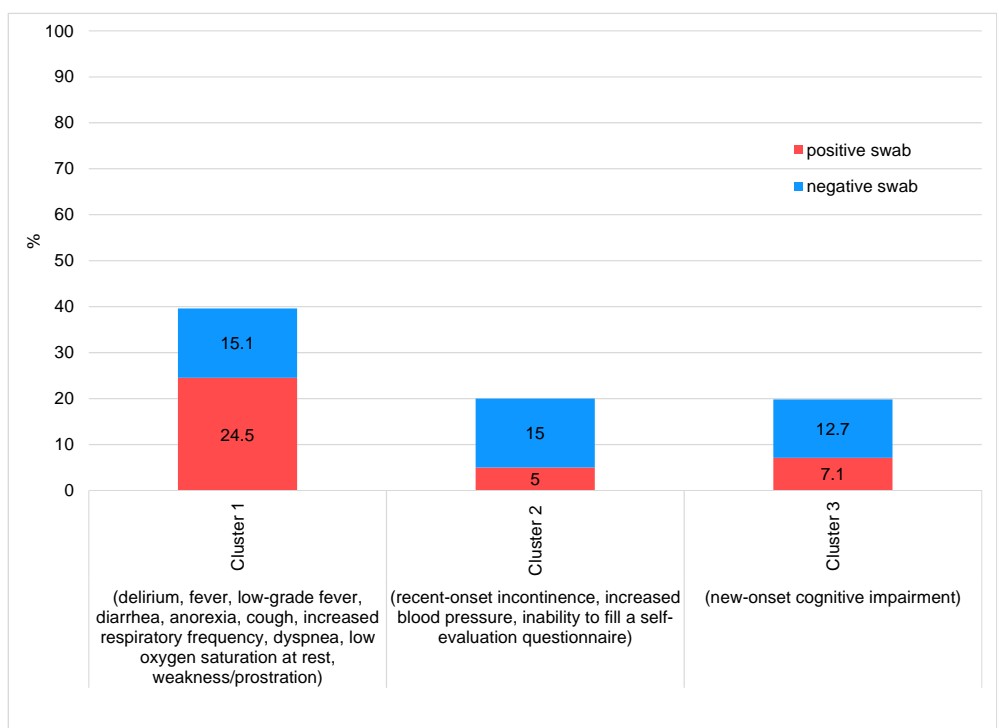

**Figure 3.** Participants (%) in each cluster and SARS-CoV-2 swab test results.

Residents were observed for a median time of 61 days, and those with positive swab test had a median duration of RT-PCR positivity for 20 days. At the end of the observation period, 71.4% of SARS-CoV-2-positive residents who had only one symptom and 47% of those with two or more symptoms showed clinical improvement. No statistical differences were found between SARS-CoV-2 positive and negative residents according to outcomes in all-cause hospitalizations and transfers to different care settings (8% vs 2% respectively; *p* = 0.115). Mortality probability rates were 19.6% and 9.6% in SARS-CoV-2 positive and negative residents, respectively. Cox regression analysis adjusted for sex and age showed that SARS-CoV-2 positivity (HR = 2.6, 95%CI: 1.8–4.6, *p* < 0.001) and the number of comorbidities (HR = 1.3, 95%CI: 1.1–1.3, *p* < 0.001) were significantly associated with a higher risk of mortality. In additional Cox regression analyses including symptom clusters, we found that Cluster 2 was not related to mortality (HR = 0.82, 95%CI: 0.26–2.56, *p* = 0.730), while Cluster 3 was significantly associated with an increased mortality rate (HR = 5.41, 95%CI: 1.56–18.8, *p* = 0.008). All deceased SARS-CoV-2 positive residents had symptoms found in Cluster 1.

## 4. Discussion

Our findings confirm recent reports underlining that the clinical presentation of SARS-CoV-2 infection in older residents differs from those in young and adult individuals [11–13]. Interestingly, we found that older residents often presented atypical and nonspecific symptoms, such as delirium, low-grade fever, and anorexia and were less likely to have dyspnea, ageusia, and anosmia. Indeed, the high prevalence of asymptomatic cases complicates infection identification, control, and the containment of new SARS-CoV-2 infections in LTCFs. Standardized assessments of single or multiple atypical signs and/or symptoms should quickly prompt COVID-19 testing in older LTCF residents. Although high- and low-grade fevers were common findings in our study, fever was not invariably present among residents with a SARS-CoV-2 infection. High-grade fever added specificity compared with low-grade fever for COVID-19 screening in our sample (high-grade fever in 39.1% of residents vs. low-grade fever in 36.2%). According to a previous observation [14], most residents had body temperature elevations when infected with SARS-CoV-2, but

rarely over 38.0 °C. Indeed, low-grade fever should be suspicious of an infection due to COVID-19 in older LTCF residents.

Delirium was the most prevalent onset symptom for a SARS-CoV-2 infection. Our finding parallels those of previous studies underlining that environmental and iatrogenic factors such as immobility, social distancing, use of sedative drugs, and quarantine increased the risk of delirium in older people in acute care [15] and in home-dwelling adults with dementia [16]. A literature analysis reported that mechanisms related to hypoxia, inflammation, and hypercoagulability in cerebrovascular events could explain the correlation between SARS-CoV-2 infection and neurological manifestations [17,18]. Along with delirium, anorexia was another prevalent symptom in our cohort and confirmed to be an important indicator of acute illness, including SARS-CoV-2 infection, in nursing home residents [19]. Interestingly, we found that hypoactive delirium in association with low- or high-grade fever, anorexia, and diarrhea was highly predictive of a SARS-CoV-2 infection. Deceased residents with a SARS-CoV-2 infection experienced more symptoms found in Cluster 1 compared with survivors. Moreover, multimorbidity (≥3 conditions) and dementia were significantly associated with SARS-CoV-2 positivity and mortality among residents. We also found that mortality not related to SARS-CoV-2 was significantly higher in residents with a negative RT-PCR test who presented with symptoms suspicious of infection [4]. Similarly, the probability of death was significantly increased in those with new-onset cognitive impairment. This finding suggests a potential role for the use of using standardized screening tools to measure cognitive impairment and delirium in SARS-CoV-2-infected LTCF residents. Future prospective studies are needed to provide important knowledge to this topic.

An additional finding from our study lies in the remarkable number of asymptomatic or paucisymptomatic residents who remained positive for several weeks. At the time of analysis, infected residents underwent prolonged periods of social isolation with negative consequences for psychological and functional status. At the moment, the European Centre for Disease Prevention and Control recommends that isolation can be lifted if one of the following applies: fever no longer present for at least 3 days, symptoms other than fever improved 20 days after the onset of symptoms, or 2 consecutive negative SARS-CoV-2 RT-PCR tests obtained in a 24-h interval from respiratory specimens [20,21]. Although the disease prognosis remains difficult to predict, most residents with a SARS-CoV-2 infection can be treated directly in LTCFs. Expertise in geriatric medicine along with appropriate staff and resources (including PPE and testing capacity) and health care policy support should be implemented in order to manage LTFS populations [22].

LTCFs in Italy have shown progressive reductions in severe SARS-CoV-2 cases, isolations, hospitalizations, and deaths since February 2021, which may be explained by the large implementation of anti-SARS-CoV-2 vaccinations [23]. Despite these encouraging results, outbreaks of COVID-19 among older vaccinated adults continue both in Italy and worldwide. Therefore, knowledge of symptoms and cluster-onset symptoms may assist in early identifying a SARS-CoV-2 infection, especially in older LTCF residents [24].

Due to the observational nature of our study, we cannot identify any cause–effect relationships between SARS-CoV-2 infection and mortality. However, our study provides an important basis for future prospective studies on specific clusters related to higher mortality rates. This research was conducted in LTCF settings, and thus, our findings may not be applicable in all types of care settings. An important strength of our study was the use of an electronic registry dedicated to precise clinical data collection in multiple settings of older persons during the first pandemic wave.

## 5. Conclusions

LTCF residents commonly present an asymptomatic or paucisymptomatic form of SARS-CoV-2 infection. In symptomatic patients, we found that key SARS-CoV-2 infection symptoms included delirium and fever (including low-grade fever), alone or in clusters. Therefore, these symptoms should be considered in early identifying and potentially

predicting the prognosis of SARS-CoV-2 infection in LTCF residents. Due to the highly contagious risk of SARS-CoV-2 spread in LTCFs, early recognition of an atypical COVID-19 presentation is pivotal.

**Author Contributions:** Study concept and design: A.M., R.A.I. and S.D.S.; Methodology: A.M., R.A.I., C.T. and S.D.S.; Software: S.D.S. and G.Z.; Investigation and Data Curation: GeroCovid LTCFs Working Group and G.B. (Gilda Borselli); Analysis and interpretation of data: A.M., M.N., A.M.A., C.T. and R.A.I.; Writing-Original Draft Preparation: A.M., R.A.I., M.N., A.M.A. and C.T.; Writing-Review & Editing, Visualization and Supervision: R.A.I., A.M., M.N., P.G., A.M.A. and G.B. (Gilda Borselli), C.C., S.F., E.M., C.T., S.V., A.C. and G.B. (Giuseppe Bellelli), F.M., C.O., S.D.S. and G.Z.; Project Administration: R.A.I. All authors have read and agreed to the published version of the manuscript.

**Funding:** This research received no external funding.

**Institutional Review Board Statement:** The study was conducted in accordance with the Declaration of Helsinki, and approved by the Ethics Committee of Campus Bio-Medico University Ethical Committee (protocol code: 22/20 OSS; date of approval: 3 April 2020).

**Informed Consent Statement:** Informed consent was obtained from all subjects involved in the study.

**Data Availability Statement:** Data are available on the Blue Companion national de-identified clinical data electronic registry.

**Acknowledgments:** The **GeroCovid LTCFs Group** members are as follows (in alphabetical order). Angela Marie Abbatecola, (*RSA INI Città Bianca, Veroli (FR)*), Domenico Andrieri, (*RSA Villa Santo Stefano, S. Stefano di Rogliano (CS)*), Rachele Antognoli, (*RSA Villa Isabella, Pisa*), Paola Bianchi, (*Associazione Nazionale Strutture Territoriali e per la Terza Età, Roma*), Carmine Cafariello, (*RSA Villa Sacra Famiglia, IHG, Roma; I RSA Geriatria, IHG, Guidonia (RM); III RSA Geriatria, IHG, Guidonia (RM); RSA Estensiva, IHG, Guidonia (RM); RSA Intensiva, IHG, Guidonia (RM)*), Valeria Calsolaro, (*RSA Villa Isabella, Pisa*), Francesco Antonio Campagna, (*Centro di Riabilitazione San Domenico, Lamezia Terme (CZ)*), Sebastiano Capurso, (*RSA Bellosguardo, Civitavecchia (RM)*), Silvia Carino, (*RSA San Domenico, Lamezia Terme (CZ); Centro di Riabilitazione San Domenico, Lamezia Terme (CZ); RSA Villa Elisabetta, Cortale (CZ); Casa Protetta Madonna del Rosario, Lamezia Terme (CZ)*), Manuela Castelli, (*ASP Golgi Redaelli, Istituto Geriatrico Camillo Golgi, Abbiategrasso (MI)*), Arcangelo Ceretti, (*ASP Golgi Redaelli, Istituto Geriatrico Camillo Golgi, Abbiategrasso (MI)*), Mauro Colombo, (*ASP Golgi Redaelli, Istituto Geriatrico Camillo Golgi, Abbiategrasso (MI)*), Antonella Crispino, (*RSA Villa Santo Stefano, S. Stefano di Rogliano (CS); RSA Villa Silvia, Altilia Grimaldi (CS)*), Roberta Cucunato, (*RSA Villa Santo Stefano, S. Stefano di Rogliano (CS); RSA Villa Silvia, Altilia Grimaldi (CS)*), Ferdinando D'Amico, (*RSA San Giovanni di Dio, Patti (ME); RSA Sant'Angelo di Brolo (ME)*), Annalaura Dell'Armi, (*III RSA Geriatria, IHG, Guidonia (RM)*), Christian Ferro, (*RSA Sant'Angelo di Brolo (ME)*), Serafina Fiorillo, (*RSA Madonna delle Grazie, Filadelfia (VV); Casa di Riposo Mons. Francesco Luzzi, Acquaro (VV); Casa di Riposo Villa Betania, Mileto (VV); Casa di Riposo Pietro Rosano, Dasà (VV); Casa di Riposo Serena Diocesi, Mileto (VV); Alloggio per Anziani Villa Amedeo, Francavilla Angitola (VV); Casa Albergo Villa Fabiola, Monterosso Calabro (VV); Casa di Riposo Villa Sara, San Nicola da Crissa (VV); Casa di Riposo Don Mottola, Tropea (VV); Casa di Riposo San Francesco, Soriano Calabro (VV); RSA Anziani, Soriano Calabro (VV); Casa di Riposo Suore Missionarie del Catechismo, Pizzo (VV)*), Pier Paolo Gasbarri, (*Associazione Nazionale Strutture Territoriali e per la Terza Età, Roma*), Roberta Granata, (*RSA Villa Sacra Famiglia, IHG, Roma*), Nadia Grillo, (*RSA San Domenico, Lamezia Terme (CZ); Casa di Riposo San Domenico, Lamezia Terme (CZ); RSA Villa Elisabetta, Cortale (CZ)*), Antonio Guaita, (*ASP Golgi Redaelli, Istituto Geriatrico Camillo Golgi, Abbiategrasso (MI)*), Marilena Iarrera, (*RSA Sant'Angelo di Brolo (ME)*), Valerio Alex Ippolito, (*Casa Protetta Villa Azzurra, Roseto Capo Spulico (CS)*), Alba Malara, (*RSA San Domenico, Lamezia Terme (CZ); Casa di Riposo Villa Marinella, Amantea (CS); Casa Protetta Madonna del Rosario, Lamezia Terme (CZ); Casa Protetta Villa Azzurra, Roseto Capo Spulico (CS); Centro di Riabilitazione San Domenico, Lamezia Terme (CZ); RSA Casa Amica, Fossato Serralta (CZ); RSA La Quiete, Castiglione Cosetino (CS); RSA San Domenico, Lamezia Terme (CZ); RSA Villa Elisabetta, Cortale (CZ); RSA Villa Santo Stefano, S. Stefano di Rogliano (CS); RSA Villa Silvia, Altilia Grimaldi (CS)*), Irene Mancuso, (*RSA San Giovanni di Dio, Patti (ME)*), Eleonora Marelli, (*ASP Golgi Redaelli, Istituto Geriatrico Camillo Golgi, Abbiategrasso (MI)*), Paolo Moneti, (*RSA Villa Gisella, Firenze*), Fabio Monzani, (*RSA Villa Isabella, Pisa*), Marianna Noale, (*RSA AltaVita, Istituzioni Riunite di Assistenza, Padova*), Mariasara Osso, (*RSA La Quiete, Castiglione Cosentino (CS)*), Agostino Perri, (*RSA*

*La Quiete, Castiglione Cosentino (CS)*), Maria Perticone, (*Casa di Riposo Villa Marinella, Amantea (CS)*), Francesco Raffaele Addamo, (*RSA San Giovanni di Dio, Patti (ME)*), Giovanni Sgrò, (*RSA Istituto Santa Maria del Soccorso, Serrastretta (CZ); RSA San Vito Hospital, San Vito sullo Jonio (CZ); Casa Protetta Villa Mariolina, Montauro (CZ); Casa Protetta Villa Sant'Elia, Marcellinara (CZ)*), Federica Sirianni, (*Casa di Riposo Villa Marinella, Amantea (CS)*), Deborah Spaccaferro, (*RSA Estensiva, IHG, Guidonia (RM); RSA Intensiva, IHG, Guidonia (RM)*), Fausto Spadea, (*RSA Casa Amica, Fossato Serralta (CZ)*), Rita Ursino, (*I RSA Geriatria, IHG, Guidonia (RM)*).

**Conflicts of Interest:** The authors have no conflict of interest to declare.

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
