# Peer review of "COVID-19 Signs and Symptom Clusters in Long-Term Care Facility Residents: Data from the GeroCovid Observational Study"

_reports, doi:10.3390/reports5030030_

Round 1

Reviewer 1 Report

This clinical study presents associations of signs/symptoms and mortality with SARS-CoV-2 infection in long term care facilities (LCTF) residents. The authors have performed clustering analysis and identified 3 major clusters based on symptoms to predict prognosis of SARS-CoV-2 infection. They conclude that symptoms of delirium, fever and low-grade fever, presenting alone or in clusters should be considered to identify and predict the prognosis of SARS-CoV-2 infection. This is a well written manuscript with novel association findings and adds value to the field of on going clinical research on Geriatric observational studies.

Major comments: 

1. Table 3: The authors are highly recommended to present their findings on association between clinical features and positive SARS-CoV2 swab test in graphical format (XY with 95% CI and r-square)

2. Figure 1: The major finding in this manuscript is identification of three symptom clusters. Although the dendogram presentation is important and acceptable, the authors are suggested to include an additional figure (pie chart preferably) that clearly shows the % of patients in each cluster along with % of patients with SARS-CoV2 infection. This will greatly add value to the manuscript and make it easier for the readers to comprehend.

Author Response

  1. Table 3: The authors are highly recommended to present their findings on association between clinical features and positive SARS-CoV2 swab test in graphical format (XY with 95% CI and r-square)

Reply: Thank you for this suggestion. We have now included a new Figure 1 as suggested and calculated Tjur R2 as described in Tjur T. Coefficients of determination in logistic regression models – A new proposal: the coefficient of discrimination. The American Statistician. 2009;366-72. doi.org/10.1198/tast.2009.08210.

  1. Figure 1: The major finding in this manuscript is identification of three symptom clusters. Although the dendogram presentation is important and acceptable, the authors are suggested to include an additional figure (pie chart preferably) that clearly shows the % of patients in each cluster along with % of patients with SARS-CoV2 infection. This will greatly add value to the manuscript and make it easier for the readers to comprehend.

Reply: We have now added a new figure (Figure 3) in the revised manuscript, showing the percentage of residents with symptoms included in each cluster, as well as the percentage of patients according to SARS-CoV-2 infection. We preferred to prepare a histogram, rather than a pie chart, considering that cluster membership is not exclusive.

Reviewer 2 Report

This study investigates the signs/symptoms (and their clustering) and mortality associated with SARS-CoV-2 infection in Long Term Care Facilities (LTCFs) residents. 

This is a cohort study on 586 aged ≥ 60-year-old residents at risk of or affected with Covid-19, enrolled from the GeroCovid LTCFs network. The Covid-19 signs/symptoms clusters were identified using cluster analysis, and their association with SARS-CoV-2 infection and mortality was evaluated using logistic regression and Cox proportional hazard models. Results: The cluster 1 symptoms (delirium, fever, low-grade fever, diarrhea, anorexia, cough, increased respiratory rate, sudden deterioration in health conditions, dyspnea, oxygen saturation, and weakness) affected 39.6% of residents, and were associated with a swab positivity (OR=7.21, 95%CI 4.78-10.80; p<.001) and were present in all deceased Covid-19 residents. Cluster 2 (increased blood pressure, sphincter incontinence) and cluster 3 (new-onset cognitive impairment) affected 20% and 19.8% of residents, respectively. The cluster 3 symptoms were associated with increased mortality (HR=5.41, 95%CI 1.56-18.8; p=.008), unlike those of cluster 2 (HR=0.82, 95%CI 0.26-2.56; p=.730). 

In LTCFs patients, delirium, fever, and low-grade fever, presenting alone or in clusters should be considered to identify and predict the prognosis of SARS-CoV-2 infection. 

In the current historical period, in which we are witnessing a new wave of the virus, it is evident that the symptomatic manifestations are not always the same, especially in a patient population such as the nursing home population.

It is therefore of great importance to identify which symptoms are most likely to correlate in elderly patients to suspect coronavirus-19 positivity, or in elderly individuals who are already virus-positive to identify those symptoms that are prognostically negative compared to others, to intervene more effectively and promptly.

the article is well structured and transparent in its theory and so are the examples are given in support of the thesis.

The level of scientific English vocabulary is within limits and is clear to the reader.

I recommend documenting, reading and if possible drawing inspiration for possible citation through the following related Covid articles:

10.1007/s42399-020-00527-2

10.1007/s42399-020-00418-6

I consider the article publishable in this journal with the changes and additions requested.

Author Response

We thank the reviewer for his/her appreciation of our work. We have now revised the discussion section with the suggested papers.

Reviewer 2:

This study investigates the signs/symptoms (and their clustering) and mortality associated with SARS-CoV-2 infection in Long Term Care Facilities (LTCFs) residents. 

This is a cohort study on 586 aged ≥ 60-year-old residents at risk of or affected with Covid-19, enrolled from the GeroCovid LTCFs network. The Covid-19 signs/symptoms clusters were identified using cluster analysis, and their association with SARS-CoV-2 infection and mortality was evaluated using logistic regression and Cox proportional hazard models. Results: The cluster 1 symptoms (delirium, fever, low-grade fever, diarrhea, anorexia, cough, increased respiratory rate, sudden deterioration in health conditions, dyspnea, oxygen saturation, and weakness) affected 39.6% of residents, and were associated with a swab positivity (OR=7.21, 95%CI 4.78-10.80; p<.001) and were present in all deceased Covid-19 residents. Cluster 2 (increased blood pressure, sphincter incontinence) and cluster 3 (new-onset cognitive impairment) affected 20% and 19.8% of residents, respectively. The cluster 3 symptoms were associated with increased mortality (HR=5.41, 95%CI 1.56-18.8; p=.008), unlike those of cluster 2 (HR=0.82, 95%CI 0.26-2.56; p=.730). 

In LTCFs patients, delirium, fever, and low-grade fever, presenting alone or in clusters should be considered to identify and predict the prognosis of SARS-CoV-2 infection. 

In the current historical period, in which we are witnessing a new wave of the virus, it is evident that the symptomatic manifestations are not always the same, especially in a patient population such as the nursing home population.

It is therefore of great importance to identify which symptoms are most likely to correlate in elderly patients to suspect coronavirus-19 positivity, or in elderly individuals who are already virus-positive to identify those symptoms that are prognostically negative compared to others, to intervene more effectively and promptly.

the article is well structured and transparent in its theory and so are the examples are given in support of the thesis.

The level of scientific English vocabulary is within limits and is clear to the reader.

I recommend documenting, reading and if possible drawing inspiration for possible citation through the following related Covid articles:

10.1007/s42399-020-00527-2

10.1007/s42399-020-00418-6

I consider the article publishable in this journal with the changes and additions requested.

Reply:  We thank the reviewer for his/her appreciation of our work. We have now revised the discussion section with the suggested papers.